# An Overview on the Conservative Management of Endometriosis from a Naturopathic Perspective: Phytochemicals and Medicinal Plants

**DOI:** 10.3390/plants10030587

**Published:** 2021-03-20

**Authors:** Andreea Balan, Marius Alexandru Moga, Lorena Dima, Catalina Georgeta Dinu, Carmen Constantina Martinescu, Diana Elena Panait, Claudia Alexandrina Irimie, Costin Vlad Anastasiu

**Affiliations:** 1Department of Medical and Surgical Specialities, Faculty of Medicine, Transilvania University, 500019 Brasov, Romania; andreea.balan@unitbv.ro (A.B.); mogas@unitbv.ro (M.A.M.); costin.anastasiu@unitbv.ro (C.V.A.); 2Department of Fundamental, Prophylactical and Clinical Disciplines, Faculty of Medicine, Transilvania University, 500019 Brasov, Romania; lorena.dima@unitbv.ro (L.D.); carmen.martinescu@unitbv.ro (C.C.M.); claudia.deliu@unitbv.ro (C.A.I.); 3Department of Law, Faculty of Law, Transilvania University, 500019 Brasov, Romania; catalina.matei@unitbv.ro

**Keywords:** endometriosis, traditional Chinese Medicine, medicinal herbs, nutraceuticals, phytochemicals

## Abstract

Background: Endometriosis is a chronic and debilitating disease, which affects millions of young women worldwide. Although medicine has incontestably evolved in the last years, there is no common ground regarding the early and accurate diagnosis of this condition, its pathogenic mechanisms, and curative treatment. Even though the spontaneous resolution of endometriosis is sometimes possible, recent reports suggested that it can be a progressive condition. It can associate chronic pelvic pain, vaginal bleeding, infertility, or malignant degenerescence. Conventional treatments could produce many side effects, and despite treatment, the symptoms may reappear. In recent years, experimental evidence suggested that plant-based medicine could exert beneficial effects on endometriosis and endometriosis-related symptoms. This study aims to highlight the pharmaceutical activity of phytochemicals and medicinal plants against endometriosis and to provide a source of information regarding the alternative treatment of this condition. Methods: For this review, we performed a research using PubMed, GoogleScholar, and CrossRef databases. We selected the articles published between January 2000 and July 2020, written in English. Results: We found 17 medicinal plants and 13 phytochemicals, which have demonstrated their beneficial effects against endometriosis. Several of their biological activities consist of antiangiogenic, anti-inflammatory effects, and oxidative-stress reduction. Conclusion: Medicinal herbs and their bioactive compounds exhibit antiangiogenic, antioxidant, sedative and pain-alleviating properties and the effects recorded until now encourage their use for the conservative management of endometriosis.

## 1. Introduction

Endometriosis is a chronic gynecological condition, which usually affects women at reproductive age and has an extremely rare occurrence after menopause [1,2]. The incidence is increasing worldwide, presently rising close to 15% [3]. Defined by the presence and proliferation of functional endometrial-like tissues in sites other than the uterus, it causes chronic pelvic pain that does not necessarily correlate with the depth or extent of tissue infiltration [4]. Other symptoms involve lumbar pain, reduced fertility or infertility, dysmenorrhea, dyspareunia, dysuria, and dyschezia [5]. Endometriotic lesions may be both endopelvic or extrapelvic. Endopelvic foci could be located into ovaries, fallopian tubes or uterosacral ligaments. Extrapelvic sites include the abdominal wall, gastrointestinal tract, urinary tract, and nasal mucosa. Some sources indicated that extrapelvic endometriosis is produced by the bone marrow-derived stem cells, which move across the peripheral circulation and generate endometriosis in obscure sites [6].

Endometriosis is considered a multi-factorial disorder of unclear etiology, induced by genetic, hormonal, and immunological factors. Several theories have been suggested in order to explain the ethiopathogenesis of endometriosis. The most accepted theory is Sampson’s retrograde menstruation theory, which implies that endometriosis develops as a result of a retrograde flow of discarded endometrial debris through the fallopian tubes towards the peritoneal cavity during menstruation [7]. Interestingly, almost 76–90% of the women with permeable fallopian tubes present retrograde menstruation, but only a small amount of them develop endometriosis [8]. Studies on non-human primate subjects showed that endometriosis can be induced by placing curetted menstrual endometrium into the peritoneal cavity [9], with a positive result in up to 100% of animals after two successive rounds of inoculation [10]. Endometriotic lesions in primates were clinically and histologically identical with the lesions found in humans [11].

The epigenetic theory suggests that during organogenesis, Homeobox and Wingless family genes are fundamental for the development of urogenital tract structures. A dysregulation at the level of Wnt/b-catenin signaling pathway will result in an abnormal arrangement of the stem cells. Pro-inflammatory peritoneal habitat, immune alteration and the abnormal arrangement of the stem cells will evolve into endometriosis [3,12]. Additionally, miRNAs dysregulations were reported as trigger factors for a more aggressive proliferation and invasion of the endometrial cells in ectopic sites. Epigenetics has an indirect role in the differentiation of bone marrow-derived stem cells by regulating the steroid action and the inflammatory microenvironment. The interaction leads to the recruitment of bone marrow-derived stem cells, which are vastly affected by the epigenetic expression [13].

The diagnosis of endometriosis is frequently delayed, sometimes up to 7–9 years. This delay is due to the unspecific symptoms and to the difficulty of pronouncing a diagnosis, which is usually established after the histological exam of the surgically removed tissues. A delayed diagnosis and treatment results in chronic pain and infertility that leads to psychological distress, hence to a poor quality of life [14].

Endometriosis is very challenging to treat. The management of endometriosis could be surgical or conservative [15]. The most commonly used therapeutic agents are nonsteroidal anti-inflammatory drugs (NSAID) and hormonal medication for repressing the ovarian function, such as oral contraceptives, gonadotropin-releasing hormone (GnRH) agonists, selective progesterone receptor modulators, androgens, or aromatase inhibitors [16]. Aromatase inhibitors, GnRH antagonists, anti-tumor necrosis factor- α (TNF-α), antiangiogenic factors and selective progesterone receptor modulators are the newest medical therapies for endometriosis [17]. These hormonal treatments have various side effects and usually they are not fully effective [18]. In addition to the drug therapy, the surgical resection is used in order to remove the detectable ectopic sites of endometriosis, leading to improved fertility, and complete or partial pain relief. However, the reoccurrence of the endometriotic lesions is still possible [19].

Natural alternative therapies consisting of medicinal plants and phytochemicals could offer new options for the management of endometriosis. Numerous natural products with different pharmaceutical effects have shown a reduction in the size of endometriotic lesions, lowered the pelvic adhesions, and improved pelvic pain and ovarian function [20]. Medicinal herbs and their bioactive compounds exhibit anti-angiogenic, anti-oxidative, sedative and pain-alleviating properties, and the beneficial effects recorded until now encourage their use for the management of endometriosis [21].

For centuries, Traditional Chinese Medicine (TCM) has attracted extensive attention for its ability to treat and alleviate a wide range of chronic diseases. Furthermore, TCM has significant advantages in treating various gynecological disorders including endometriosis [22]. Considering the complex pathogenesis of this dysfunction and the limited effects of the conventional treatments, Chinese medicines were used to treat the endometriotic lesions and to control endometriosis-related symptoms such as pelvic pain, dysmenorrhea, and abnormal uterine bleeding [20]. Considering the complexity of TCM, the differences in clinical experience and the lack of clinical studies, a depth research of the mechanisms of action of the herbs is required [23].

In this article, we focused on the studies regarding the pharmaceutical effects of medicinal plants and phytochemicals against endometriosis. Our main objective is to bring together all these studies in order to establish whether and through which mechanisms these natural remedies can relieve the symptomatology and can influence the progression of endometriotic lesions.

## 2. Materials and Methods

This study is a review of the last twenty years of English literature, regarding the alternative treatments of endometriosis. With an ongoing interest in the field of plant-based therapy for endometriosis, we aimed to address this topic given the high burden that this gynecological dysfunction possesses on public health. A long array of studies on the conventional medical treatment of endometriosis were conducted. However, plant-based therapies raised more and more attention for their pharmaceutical properties against this gynecological pathology and our objective was to bring together all the studies conducted on this topic. Our research included all the papers published during January 2000 and July 2020, related to herbal and phytochemicals therapy for endometriosis, and to their mechanisms of action. We used the following Medical Subject Headings (MeSH) keywords: “endometriosis”, “Traditional Chinese Medicine”, “medicinal plants”, “nutraceuticals”, and “phytochemicals”. Two authors separately identified the relevant papers and selected them based on the following inclusion criteria: full-text original articles, written in English, conducted in vivo, in vitro or clinical trials on humans. The exclusion criteria were: papers written in other language than English, abstracts, and duplicate papers. A total number of 143 studies were selected and after the exclusion of duplicates, 75 studies fitted our area of interest and respected our criteria.

## 3. Pathogenic Pathways of Endometriosis

### 3.1. Inflammatory Pathways in Endometriosis

Endometriosis is a complex gynecological disorder resulted from the interaction of various hormonal, immunological, and genetic factors. It is especially characterized by chronic inflammation and increased angiogenesis [24]. The local inflammatory response represents the main pathway for the initiation and progression of the disease [25]. In healthy women, the peritoneal fluid contains almost 85% macrophages [26]. Many studies revealed that the peritoneal fluid of the affected women contains increased levels of immune cells and macrophages, leading to an increased secretion of prostaglandins, cytokines, growth factors, and angiogenic factors, such as TNF-α, interleukin-1 (IL-1), interleukin-8 (IL-8), transforming growth factor beta (TGF-β), and interferon-γ [27].

Prostaglandin E2 (PGE2) is an eicosanoid with various physiological and pathological functions that have been considered indispensable for the development of endometriosis. According to Wu et al. [28], PGE2 regulates cell proliferation, angiogenesis, immune suppression, and it represents a crucial point in the molecular mechanism of endometriosis [29]. PGE2 interacts with its receptors, known as EP1, EP2, EP3, and EP4, and acts on specific target cells through alternate or opposing intracellular pathways [30].

The inflammatory environment of this disease points out an increased production of estrogens. This event will raise the production of PGs through the activation of both Nuclear factor κB (NF-kB) and cyclooxygenase-2 (COX-2) [31]. In ectopic endometrial areas, the levels of COX-2 are very increased. PGE2 increases the levels of COX-2 in both ectopic and eutopic endometrium. A paper of Banu et al. [32] highlighted the influence of COX-2 and PGE2 on the promotion of the migration and invasion of endometrial cells. Furthermore, the inhibition of COX-2 was able to decline the invasion of epithelial and stromal cells in endometriosis.

Eutopic endometrial cells are rich in NF-k subunits which activate during menstruation. The activation of these subunits in endometriotic cells in vitro increased the secretion of proinflammatory cytokines, IL-6, IL-8, ICAM-1, granulocyte-macrophage colony-stimulating factor (GM-CSF), TNF-α, macrophage migration inhibitory factor (MIF), and MCP-1 [33]. The inhibition of this molecule in nude mice with early-stage endometriosis decreased the proliferation of endometriotic cells and stimulated apoptosis in both stromal and epithelial cells [34]. The common point of view of the reaserchers is that in endometriosis, NF-kappa B represents the link between the inflammation process and the expression of aromatase. The activation of NF-kappa B subunits, followed by the translocation from the cytoplasm to cell nuclei, represents the first step to induce the inflammatory process [35]. Activated NF-kB will release IL-6, IL-8, IL-1, TNF- α, and VEGF, leading to chronic inflammation.

Immune dysfunction characterized by hyperactive peritoneal macrophages with altered phagocytic ability represents another key-point in endometriosis development and progression. The phagocytic ability of macrophages is mediated by the secretion of matrix metalloproteinases (MMP). These are enzymes capable to destroy the organization of extracellular matrix, and the expression of several macrophages surface receptors. MMPs are also able to promote the dissolution of the cellular debris [36].

Different alterations of the MMPs activity are considered compelling factors in endometriosis development [37,38]. Most of the MMPs are secreted as dormant pro-enzymes, which later will be activated via proteolytic action. MMPs action is controlled by tissue inhibitors of metalloproteinases (TIMPs), and by the endogenous inhibitors [39]. Recent studies reported an increased expression of MMP-9, -7, -3, and -2 in patients with endometriosis [40,41]. Elevated levels of MMP-3 have been observed in animal models with induced endometriosis [42]. MMP-2, also known as gelatinase A, promotes tumoral cells migration though the degradation of collagen X, V, and IV, which represents an important structural part of the basement membrane [43]. The peritoneal fluid of endometriosis patients contains elevated levels of MMPs-2 comparing to healthy patients [44]. Additionally, the expression of MMP-2 in the peritoneal fluid of the affected women has been positively associated with the level of 17β-estradiol, and negatively associated with serum levels of progesterone [45].

Recent reports have pointed out the major role of reactive oxygen species (ROS) in endometriosis-related inflammation. In healthy individuals, there is a balance between ROS and antioxidants. When this balance bends toward ROS overproduction, oxidative stress increases, leading to the inflammatory process via the upregulation of various proinflammatory factors [46,47]. Hydroxyl radicals are considered to be the highest reactive free radical species [48]. They have the capacity to react with many cellular constituents such as amino acid residues, and also to attack the annular lipid shell, leading to lipid peroxidation [48,49]. The excessive production of ROS causes cellular damage and alters the function of the cells by controlling gene expression and proteins activity [50,51]. It has been demonstrated that ROS regulates the nuclear factor κB (NF-κB), which has been associated with endometriosis [52]. NF-κB promotes the expression of several genes that encode proinflammatory cytokines, adhesion molecules, angiogenic and growth factors, and cyclooxigenases [25,52]. This process leads to macrophages activation, endometrial cells proliferation, and increased adhesion and neovascularization [53]. The essential role of ROS in endometriosis consists of the positive regulation of transcription factors such as NF-kB and activator-protein 1. These molecules regulate several genes involved in the cellular defense and in the expression of various proinflammatory cytokines, integrins and cadherins [54].

### 3.2. Angiogenesis in Endometriosis

Angiogenesis is a biological process, which consists of the formation of novel blood vessels, increased secretion of growth factors, dissolution of the extracellular matrix, multiplication, and agglutination of the endothelial cells in order to form new capillaries-like tubes [55].

VEGF is an angiogenic growth factor, usually highly expressed in tumors and endometriosis. This molecule increases the permeability of endothelial cells via two mechanisms: by enhancing the activity of vesicular-vacuolar organelles and by mitogen-activated protein (MAP) kinase signal transduction cascade, which targets the loose of adhering junctions between endothelial cells via rearrangement of adhesion complexes [56,57]. VEGF is a mitogen with high specificity for endothelial cells, which promotes their proliferation after binding its receptors: Flt-1/VEGFR-1, and Flk-1/KDR/VEGFR-2 [58].

VEGF exhibits angiogenic effects by inducing the expression of several molecules, such as α1β1, α2β1, and αvβ3-integrins. These integrins regulate and promote cellular migration, proliferation, and matrix reorganization [59]. The transcription of VEGF mRNA, induced by hypoxia, is regulated through a mechanism, which consist of the binding of hypoxia-inducible factor 1 (HIF-1) to an HIF-1 binding site. This site is located in the promoter of VEGF. The transcription of VEGF mRNA is triggered by the activation of a stress-inducible PI3K/Akt pathway [59].

Fibroblast growth factors (FGFs) levels are increased in cancers and endometriosis, because these pathologies involve increased rates of angiogenesis. FGFs are important angiogenic molecules and endothelial cells survival factors. In addition, the expression of VEGF mRNA in endometriosis is up-regulated by high levels of bFGF. Overexpression of this molecule also up-regulates the expression of the proteins Bcl-XL and Bcl-2, with anti-apoptosis role, through the MEK/ERK signaling pathway [60].

In the last years, accumulating evidence showed that miRNAs target various angiogenic factors and protein kinases, and enhances neovascularization. In addition, these molecules can promote or inhibit angiogenesis in endometriotic lesions, through the modulation of pro-angiogenic signals (induced by VEGF) or anti-angiogenic signals (induced by thrombospondin-1 (TSP-1)). Moreover, miRNA targets receptor tyrosine kinases (RTKs) and hypoxia-inducible factor (HIF) and can crosstalk with ROS, influencing the angiogenesis process [61].

A study conducted in 2010 by Lee and coworkers [62] pointed out that the proangiogenic factor prokinetitsina-1 (PK-1) is highly expressed in ectopic foci of endometrial cells in comparison with eutopic cells.

### 3.3. Apoptosis and Endometriosis

Apoptosis is described as a physiological process, which induces serious cellular modifications, including pyknosis, nuclear fragmentation, blebbing of the plasma membrane, and cell shrinkage [63]. The apoptosis process ends with the production of small cellular fragments, known as apoptotic bodies. Apoptotic bodies are removed by phagocytosis, without enhancing a pro-inflammatory response [64]. In endometriosis, endometrial cells possess the ability to avoid apoptosis when they are localized in ectopic areas.

The apoptotic process is triggered by caspases, a group of protease enzymes, which determine the cleavage of C-terminal fragments into residues of aspartic acid upon activation [65]. The main class of caspases is represented by caspase-9 homologues, also called “effector caspases” or “initiators”. In animals, a complex composed by cytochromes, CED-4/Apaf-1 and procaspase-9, known as apoptosome, determine the activation of procaspase-9, which causes the effector caspase downstream cascade, and consequently, the degradation of the cells [66]. It is considered that apoptosis is triggered via extrinsic pathway, through the interaction between TNFα, Fas ligand (FasL/CD95L), transforming growth factor β (TGFβ), caspases and B cell lymphoma-2 (Bcl-2) family. Different studies pointed out that the endometrium of the endometriotic women expresses lower levels of pro-apoptotic factors (e.g., Bax), and increased levels of Bcl-2, an anti-apoptotic factor, in comparison with unaffected women [67].

Recent studies have shown that microRNA (miRNA) dysregulations play a role in endometriosis development [68]. The analysis of miR-183 revealed its involvement in the induction of apoptosis in human endometrial cells. A decreased expression of miR-183 associated with endometriosis, leads to decreased apoptosis and, afterwards, to the advancement of the disease [69].

Existing data suggests that miR-191 suppresses TNFα-induced cell death in ectopic endometrium from endometriosis patients. Additionally, miR-191 targets DAPK1, which represents a mediator of the cellular death. Thus, miR-191-DAPK1 pathway could not only be involved in the development of endometriotic lesions, but it could determine the malignant transformation of this condition [70].

In our opinion, knowing the molecular signaling pathways in endometriosis is essential for choosing the most appropriate therapeutic strategies to conquer this chronic enigmatic gynecological disease.

## 4. Results

### 4.1. Phytochemicals for the Treatment of Endometriosis

As we previous mentioned, the conventional treatment for endometriosis is still limited and associated with side effects. For this reason, an increasing number of women search for alternative options, such as phytotherapy, which is a promising strategy for the management of endometriosis. Phytochemicals such as flavonoids or phenolic acids proved their beneficial effects by exerting antioxidant, anti-inflammatory, immunomodulatory, and pro-apoptotic functions [71]. However, these treatments are still unconventional and their administration should be discussed in advance with a specialist.

#### 4.1.1. Apigenin

Apigenin is the most active flavone which can be found in dietary sources such as chamomile tea, parsley, celery, oranges, celeriac, or wheat [72]. It possesses various biological activities such as antiproliferative, anti-angiogenic, anti-oxidative, and anti-inflammatory properties. Adding apigenin to endometriotic stromal cells obtained after surgery, Suou et al. [73] observed that it attenuated the protein expression and TNF-α-induced IL-8 gene expression. Two years later, same authors showed that apigenin can suppress cellular proliferation induced by TNF-α. It attenuated chronic inflammation and mitogenic activity by suppressing PGE2 expression and by downregulating the NFkB pathway [74].

Park et al. [75] used human endometriosis cell lines (VK2/E6E7 and End1/E6E7) and observed that apigenin induced apoptosis and cell cycle arrest. Furthermore, it disrupted the mitochondrial membrane potential and increased cytosolic concentration of Bax and cytochrome c, which are pro-apoptotic proteins. A recent study [76] pointed out that apigenin is a potential phytoprogestin, a natural progesterone-like molecule capable to interact with the progesterone receptor (PR). It blocked the increase of endometrial cells height, suppressed genistein-induced proliferation of the endometrial cells and increased the expression of Hand2, a transcription factor stimulated by PR. In conclusion, apigenin is valuable for its antiproliferative, anti-angiogenic, anti-oxidative and anti-inflammatory effects, but it is also useful for its progesterone-like activity.

#### 4.1.2. β-Caryophyllene

β-caryophyllene is a natural sesquiterpene, also known as phytocannabinoid. It can be found in essential oils of different medicinal or ornamental plants, spices or fruits. Food and Drug Administration approved this phytochemical as flavoring agent and taste enhancer [77]. Recently, it has been reported that β-caryophyllene can suppress tumoral growth and metastasis, and can induce apoptosis. These biological effects are possible through the suppression of PI3K/AKT/mTOR/S6K1 signaling pathways and the activation of ROS-mediated MAPKs. β-caryophyllene potentiates apoptosis and suppresses tumoral cells invasion via the activation of NF-kB. Furthermore, it has been demonstrated that β-caryophyllene suppresses the expression of Bcl-xL, IAP1, IAP2, Bcl-2, VEGF, MMP-9, COX-2, ICAM-1, and c-Myc [78].

According to several researchers [79,80], β-caryophyllene also exerts potent anti-inflammatory activity in vivo, via decreasing the levels of TNF-α, TLR-4, and IL-1ß. Using a rat model, Abbas et al. [80] showed that β-caryophyllene induced apoptosis in luminal epithelium of the ovarian endometriomas and in endothelial cells. In addition, this phytochemical decreased the growth of endometriotic foci by more than 52.5% in comparison to controls. β-caryophyllene is also able to modulate the level of oxidative stress. It restores antioxidant enzymes and represses glutathione (GSH) depletion and lipid peroxidation [81].

#### 4.1.3. Curcumin

Curcumin is a hydrophobic polyphenol, also known as turmeric, derived from the rhizome of *Curcuma longa.* It possesses a wide range of pharmaceutical activities, such as anti-oxidative, anti-angiogenic, anti-proliferative, growth-suppressive, anti-inflammatory, and antimetastatic [82]. In endometriosis, curcumin reduced the number of endometriotic cells through the modulation of estrogen levels [83]. After the treatment with curcumin, the secretion of IL-1ß and NF-κB was significantly suppressed [84]. In addition, the release of TNF-α and TGF-ß1 can be modulated by curcumin. It induces apoptosis via p53 and PI3K signaling pathways and through the up-regulation of the protein YAP/p53 [85].

This polyphenol exerts its anti-angiogenic function by inhibiting the VEGF/NF-κB signaling pathways and reducing the expression of VEGF [86]. In heterotopic endometrial tissue of rats, curcumin lowered VEGF expression and reduced the quantity of microvessels [87]. Furthermore, it proved its anti-oxidative potential by enhancing superoxide dismutase (SOD) and glutathione peroxidase (GPx) levels [86]. Lee et al. highlighted that curcumin decreases the invasiveness of tumoral cells by decreasing serum levels of MMP-2 and MMP-9, and altering the p53-E-cadherin pathway [88].

Sayantan et al. [42] showed that curcumin regresses endometriosis by inhibiting NF-kB translocation and MMP-3 expression. Moreover, it is able to induce apoptosis in ovarian endometriosis via cytochrome-c mediated mitochondrial pathway, independently of p53 protein.

It has been demonstrated that curcumin also inhibits the migration and invasion potential of tumoral cells. The synergistic treatment with curcumin and extracellular signal-regulated kinase was leading to a significant decrease of invasive capabilities of tumoral endometrial cells, secondary to MMP-2 and MMP-9 reduction [89].

#### 4.1.4. Epigallocatechin-3-gallate

Epigallocatechin-3-gallate (EGCG) is extracted from *Camelia sinensis* and is the most bioactive molecule in both green and black tea. It possesses antiangiogenic, pro-apoptotic and anti-proliferative effects [90] and due to its pharmaceutical effects, it could be used in the future in the management of endometriosis. No clinical trials on humans have been conducted until present. However, the results of the experiments realized on animals provided promising perspectives.

Xu et al. [91] reported that EGCG suppressed vascular endothelial growth factor C (VEGFC) and vascular endothelial growth factor receptor 2 (VEGFR2) in endometriotic foci. Furthermore, it significantly decreased the expression of VEGFC in endothelial cells and downregulated VEGFC/VEGFR2 signaling pathway in vivo, leading to decreased angiogenesis.

Endometriosis is characterized by fibrosis. A dense fibrous tissue induced by chronic inflammation surrounds the stroma and endometrial glands. Recent animal experiments pointed out that EGCG could prevent the fibrosis progression in endometriosis [92]. In vivo studies showed that the treatment with EGCG inhibited the proliferation, migration and invasion of endometriotic cells. Moreover, it inhibited TGF-β1-stimulated activation of MAPK and Smad signaling pathways and the expression of miRNA TGF-β1-dependent [92].

Pro-EGCG is a derivative of EGCG obtained by acetylation, which showed more potent anti-angiogenic effects that EGCG in mice models. It also possesses a better bioavailability in vivo in comparison with EGCG. Both EGCG and pro-EGCG inhibit the expression of VEGF and VEGFR activity, being valuable anti-angiogenic agents. Furthermore, they exert anti-oxidant activity in vitro and in vivo [93]. EGCG decreased the oxidative stress by activating the ERK1/2 and the p38 MAPK signaling pathways. The consequence was the up-regulation of Nrf2/HO-1 and the suppression of MAPK pathway.

In plasma, EGCG reduced the level of pro-inflammatory cytokines IL-17, IL-22 and IL-23. In addition, the level of tumor necrosis factor alpha (TNF-α), IL-18, CD18, CD11s, and miRNA was decreased by EGCG administration [94,95]. EGCG is also able to promote apoptosis in endometriotic cells. Huang et al. [96] showed that EGCG promoted apoptosis in vitro by increasing the levels of caspase-3 and using STAT3 and P53/Bcl-2 signaling pathways. Moreover, it suppressed the migration and invasion of tumoral cells by inducing the down-regulation of MMP-2 and MMP-9 [97].

#### 4.1.5. Genistein

Genistein is an isoflavone isolated from soy. It possesses strong phytoestrogenic effects and both in vivo and in vitro studies have reported its efficiency for the management of endometriosis [4]. This molecule was reported to decrease the surface of the endometriotic implants and the histopathologic score through anti-angiogenic and anti-proliferative mechanisms [98]. Genistein exerts its anti-angiogenic effects by reducing the activity of PTK and MAPK, and by inhibiting HIF-1 and VEGF gene expression [99]. It induces programmed cellular death by suppressing Notch1/NF-κB /slug/E-cadherin pathway and by inducing G0/G1cell cycle arrest [100].

Genistein is also known for its antioxidant activity. The mechanism consist of the upregulation of Nrf2 and HO-1, and activation of Akt [101]. In addition, genistein was reported to decrease the expression of a wide range of pro-inflammatory factors, such as COX2, IL-6, TNF-α, NF-κB, and IL-1ß [102].

In order to sustain the potential of genistein as phytoestrogen, Cotroneo et al. [103] attached uterine tissue to intestinal mesentery of Dawley rats. They observed that injections of genistein reduced uterine ER-α, while pharmacologic injections of genistein significantly increased PR(B). They concluded that pharmacologically and not dietary genistein administration supported surgically induced endometriosis in ovariectomized rats and this molecule is a strong estrogen agonist.

#### 4.1.6. Luteolin and Chrysin

Luteolin is a flavonoid that belongs to flavones group, along with chrysin and apigenin. It can be extracted from carrots, broccoli, green pepper, chamomile tea, or parsley [104] and it possesses antiproliferative and anti-inflammatory effects. Luteolin inhibited cell proliferation by inducing cell cycle arrest and DNA fragmentation in VK2/E6E7 (vaginal mucosa-derived epithelial endometriotic cells) and End1/E6E7 (endocervix-derived endometriotic cells) endometriotic cells. In endometriosis mice, intraperitoneally injected with luteolin, a significant reduction of the lesions size was reported. Furthermore, luteolin suppresses endometriosis progression by downregulation of the MAPK and PI3K/AKT signal proteins [105].

Luteolin can be considered a potent endocrine disruptor, which display progesterone antagonist activity. It possesses strong estrogen agonist activity and can modulate the growth of estrogen-dependent cells and tissues [106].

Chrysin is a natural flavone, especially derived from bee products such as propolis, honeycomb or honey. It can also be extracted from passion flowers and chamomile [107]. There is a small amount of studies conducted in order to elucidate the biological mechanism of chrysin on endometriosis. Ryu et al. [108] showed that chrysin exerted pro-apoptotic effects on End1/E6E7 and VK2/E6E7 endometriotic cells. Furthermore, it inhibited the proliferation of the cells. The programmed cell death was induced by increasing intracytosolic level of reactive oxygen species. Moreover, chrysin downregulated PI3K/AKT signaling pathway in a dose-dependent manner.

#### 4.1.7. Myricetin

Myricetin is a natural compound included in the flavonols group. It is a plant-derived α-glucosidase and α-amylase inhibitor, with strong antioxidant activity [109]. It has been reported to be very efficient for relieving type 2 diabetes mellitus associated symptoms [109]. In a recent study, Park et al. [110] reported for the first time the antiproliferative effects of myricetin in endometriosis. They demonstrated that this compound inhibits cell cycle progression and cellular proliferation of human End1/E6E7 and VK2/E6E7 cells. In addition, myricetin leads to apoptosis by inducing the loss of mitochondrial membrane potential and ROS accumulation. In mouse model, this flavonol regressed the size of endometriotic foci by inhibiting Ccne1, and decreased the activation of AKT and ERK1/2 proteins. Considering these findings, myricetin could be a promising molecule for the management of endometriosis, but further research are urgently necessary in order to better describe the mechanisms of action against this gynecological condition.

#### 4.1.8. Naringenin

Naringenin is included in the flavanones group and can be found in citrus and grapes. Its pharmaceutical effects consist of anti-inflammatory, antimutagenic, and anticancer properties [111]. Furthermore, it has been reported that naringenin induces apoptosis in various cellular lines, but unfortunately it has poor bioavailability and low aqueous solubility [112]. Naringenin inhibits cells proliferation and exerts pro-apoptotic effects in human endometriosis cells, through the depolarization of mitochondrial membrane potential. Moreover, naringenin is able to increase intracellular levels of ROS [113].

Naringenin ameliorated the expression of VEGF, reduced the volume of the endometriosis lesions and decreased serum levels of TNF-α in rats after 21 days of administration. In addition, naringenin reduced the expression of MMP-2 and MMP-9 in in vitro cells culture, by inhibiting the invasion of these cells [114].

#### 4.1.9. Puerarin

Puerarin is an isoflavone with phytoestrogenic potential, which can be found in the roots of *Pueraria* spp. In rat models with endometriosis, was performed oral gavage with puerarin, and the results were evaluated four weeks after the administration. The weight of the ectopic endometrial tissue and the estrogen levels were significantly decreased in the study group in comparison to controls. These pieces of evidence demonstrated that puerarin is able to suppress the development and growth of endometriosis [115].

Wang et al. [116] have demonstrated that puerarin is able to suppress the vascularization and invasion of estrogen-stimulated endometriotic tissues. E2 increases MMP-9 and decreased TIMP-1 accumulation, leading to an increased invasiveness of endometriotic cells. Puerarin efficiently reversed these effects and, in addition, it degreased the levels of VEGF and ICAM-1, proving its anti-inflammatory and anti-angiogenic activities. Another study from the year 2012 [117] pointed out that puerarin significantly decreased the proliferation of endometriotic cells induced by 17ß-estradiol. The mechanism consisted of the downregulation of Cyclin D1, Cox-2, Cyp19, and by the abrogation of ERK pathway phosphorylation. Moreover, puerarin has been reported to downregulate mTOR signalling pathways, NF-kB signalling pathways, BCl-2 and PI3K proteins, and to increase the activity of c- Jun N terminal kinase, miR-16, and extracellular signal-regulated kinase by 50%. All these mechanisms lead to decreased cell proliferation and increased programmed cellular death [118], which transform puerarin into a promising therapeutic agent for the management of endometriosis.

#### 4.1.10. Quercetin

Flavonols is a class of flavonoids, which is widely distributed in various food sources such as apples, berries, broccoli, red grapes, onions, tea, and seeds [119]. Quercetin is a flavonol, which inhibits the cellular growth of cancer cell lines and induces antiproliferative effects in endometriotic cells, both in vivo and in vitro. Scambia et al. [120] showed that quercetin produced a dose-dependent inhibition of endometriotic cells. Zhang et al. evaluated the effects of quercetin on endometriosis rats. They observed that the size of endometriosis foci significantly decreased, along with a significant reduction of VEGF and heat shock protein 70 (HSP70) expression [121].

Furthermore, quercetin decreased the expression of PR and estrogen receptors (ER) α and β in hypothalamus, pituitary gland, and endometrium of the endometriosis rats. In these conditions, quercetin could be considered an antiestrogen and progesterone molecule [122].

#### 4.1.11. Resveratrol

Resveratrol is a natural phytoalexin, included in the family of stilbenes [35,123]. Various plants synthesize it in the presence of ultraviolet radiations. Wine, grapes, stilbenes, and berries represent the most common sources of resveratrol [124]. Resveratrol possesses various beneficial therapeutic effects, such as anti-inflammatory, anti-angiogenic, anti-oxidative and anti-neoplastic properties [125].

Resveratrol exerted its anti-inflammatory effects through the down-regulation of pro-inflammatory cytokines, and inhibition of COX-2 and prostaglandin expression [126]. In addition, it lowered the expression of ICAM1, suppressed MCP1, and inhibited NF-κB/TLR-4 p65/MAPKs signaling cascade [127]. The suppression of COX-2 and aromatase in ectopic endometrial areas seems to be essential for the alleviation of endometriosis-related chronic pelvic pain. In addition to this ability possessed by resveratrol, it can also block SIRT 1 and TGF-beta genes [128].

Resveratrol also possesses anti-angiogenic effects, manifested via GSK3beta/beta-catenin/TCF-dependent pathway downregulation and VEFG inhibition [129]. Furthermore, this phytochemical is a potent antioxidant. Increased production of free oxygen radicals and low levels of glutathione peroxidase (GSH-Px) and SOD characterize endometriosis. After the supplementation with resveratrol, it was reported that the increased peritoneal levels of free oxygen radicals were suppressed [130]. Moreover, resveratrol attenuated the oxidative stress via activating Nrf2/HO-1 pathway and decreasing the levels of HO-1, catalase (CAT), SOD, and GPx [131,132].

#### 4.1.12. Xanthohumol

Xanthohumol is a bioactive molecule contained by the female inflorescence of *Humulus lupulus* L., which has been widely used in plant-based medicine due to its beneficial activities. Beer is the most important dietary source of xanthohumol [133]. Due to its anti-inflammatory, anti-proliferative, and anti-angiogenic properties, this prenylated flavonoid is known as a cancer chemopreventive agent. Rudzitis-Auth et al. [134] have reported that xanthohumol decreased the size of peritoneal ectopic endometrial tissue areas in mice, and reduced the level of phosphoinositide 3-kinase protein. Moreover, this molecule selectively lowered the density of microvessels from the endometriotic foci, without affecting uterine or ovarian vascularization. The anti-angiogenic function of xanthohumol was exerted via AKT/ NF-κB inhibition [135]. Furthermore, it has been reported that xanthohumol is able to modulate the inflammatory response, by suppressing the gene expression of proinflammatory molecules, such as IL-1, IL-6, MCP-1, and ICAM-1 [136].

Figure 1 is a schematic representation of the main mechanisms of action of these phytochemicals against endometriosis.

### 4.2. Medicinal Plants for the Treatment of Endometriosis

#### 4.2.1. *Angelica sinensis* (Danggui)

*A. sinensis* is a Chinese herbal medicine used for centuries for its anti-inflammatory, antioxidative, immunomodulatory, and antitumoral effects. In gynecology it has been used to treat menstrual disorders, such as dysmenorrhea or amenorrhea [137]. Over 70 bioactive molecules have been isolated from *A. sinensis,* such as phenols, amino acids, essential oils, carbohydrates, organic acids and vitamins [138], but the major constituents remain polysaccharides [139].

Xiong et al. [140] investigated the effects of *A. sinensis* on human endometriotic cells and rats. They observed that *A. sinensis* extracts significantly decreased inflammation both in vivo and in vitro. After the treatment, the peritoneal levels of TNF-α and IL-18 registered significantly lower values and CA-125 decreased. Moreover, it suppressed the expression of MMP-2 and MMP-9.

#### 4.2.2. *Achillea biebersteinii* (Yarrow)

*A. biebersteinii* belongs to the family of Asteraceae, and in Turkish folk medicine it has been widely used for its emmenagogue potential [141]. The aerial parts of *Achillea* genus medicinal plants contain many biological active compounds such as flavonoids, monoterpenes, and sesquiterpenes [142], known for their beneficial effects consisting of anti-inflammatory, analgesic and antiseptic activities.

Demirel et al. [143] reported that *A. biebersteinii* reduced the volume of ectopic endometrial tissue areas and inhibited the developments of pelvic adhesions. After the administration of *A. biebersteinii* extracts, the levels of IL-6, VEGF, and TNF-α significantly decreased. In combination with *Foeniculum vulgare*, *Peganum harmala*, *Nigella sativa,* and *Curcumin cyminum*, this medicinal herb exerted strong effects against endometriosis-related dysmenorrhea. Furthermore, ethanol extracts of *A. biebersteinii* have been reported to possess strong antioxidant activity [144], and methanolic flower extract showed antinociceptive effects in mouse pain models, mediated by the cholinergic receptor [145].

#### 4.2.3. *Artemisia princeps*

*A. princeps* belongs to the family Asteraceae, and its leaves have been used from years for the management of dysmenorrhea, infertility, and other endometriosis-related complaints. This medicinal plant contains a wide range of bioactive compounds such as flavonoids, terpenoids, sterolic acids, and coumarins [146]. It has been reported to have antitumor, antioxidant, antispasmodic, antihemorrhagic, and antiulcerogenic effects [4].

Kim et al. [147] used human endometriotic cells and reported that *A. princeps* leaf extract induced apoptosis in vitro. Pro-apoptotic mechanism consisted of the regulation of p38 and NF-kB pathways. In addition, *A. princeps* extracts inhibited the expression of Bcl-2, Bcl-xL, XIAP, caspase 3, caspase 8, and caspase 9 in a dose-dependent manner.

Another study [148] reported antiproliferative activity of eupatilin, a naturally flavonoid extracted from *A. princeps,* against tumor endometrial cells. This compound inhibited endometrial cells growth via G2/M phase cell cycle arrest, and up-regulated p21 protein, by the inhibition of mutant p53. *A. princeps* leaf extract also inhibited the proliferation of T lymphocyte and the secretion of IFN-γ and IL-2. Ethanolic extract of this medicinal herb lowered the serum levels of TNF-α, ICAM-1, IL-1ß, and VCAM-1, according to Han et al. [149].

Jaceosidin is a natural flavone, which can be found in *A. princeps.* Kim et al. [150] pointed out that jaceosidin possesses antioxidative and anti-inflammatory activity. It suppressed RO, nitric oxide (NO), and NF-kB and lowered the expression of nitric oxide synthase (iNOS) in lipopolysaccharide (LPS)-induced macrophages.

#### 4.2.4. *Allium sativum*

*A. sativum* also known as garlic, is part of the Liliaceae family, and it is one of the most commonly used plants, with various pharmacological properties. Using human endometrial stromal cells, Kim et al. [151] showed that hexane extract of aged black garlic reduced cellular proliferation through the reduction of VCAM-1 and ICAM-1 expression. Further, pro-apoptotic activity of *A. sativum* was confirmed via caspase-3 increased activity and by Bax: Bcl-2 increased ratio.

Xiao et al. [152] demonstrated for the first time that diallyl trisulfide, a constituent of garlic, is able to inhibit angiogenesis in human endothelial cells. It suppressed the formation of novel capillaries, decreased the secretion of VEGF and VEGF receptor 2 and inactivated Akt kinase. Furthermore, *A. sativum* suppressed the secretion of pro-inflammatory cytokines such as Il-2, IL-6, Il-8, TNF-α, IFN-γ and enhanced the secretion of IL-10, an anti-inflammatory cytokine [153].

*A. sativum* also possesses antioxidative effects. Padiya et al. [154] pointed out that it reduced oxidative stress via AKT/PI3K/Nrf2-Keap1 pathway activation. In addition, garlic extracts inhibited peroxidation processes through the reduction of plasma methylenedioxyamphetamine (MDA) levels, and enhancement of various key antioxidant enzymes [155].

#### 4.2.5. *Astragalus membranaceus*

*A. membranaceus* is a Chinese medicinal herb, which contains a wide range of bioactive chemical constituents: formononetin, adenosine, saccharose, calycosin, ononin, calycosin-7-O-beta-D-glucoside, daucosterol, and 9,10-dimethoxypterocarpan-3-O-beta-D-glucoside. It has been reported as a useful anti-proliferative and antioxidant agent. Yoon-Sang et al. [156] administered *A. membranaceus* extracts to rats with surgically induced endometriosis. After 40 days, the volume of ectopic endometrial tissue areas was significantly lower. Furthermore, orally administered extracts of *A. membranaceus* decreased the concentration of estrogen, progesterone, IL-2 and TNF-α. Other findings suggested that Astragalus roots suppressed estrogen-dependent endometrial cells proliferation and infertility-related ovarian dysfunctions [157]. According to Zhao et al. [158], the treatment with *A. membranaceus* and other Chinese medicinal herbs prevented the recurrence of endometriosis after conservative surgery, increased the fertility and had fewer side effects in comparison with the conventional medical treatment.

#### 4.2.6. *Curcuma longa*

*Curcuma longa* is a medicinal herb also known as turmeric, whose major bioactive compound is represented by curcumin. Kong et al. [159] evaluated the effects of combined therapy with Tamoxifen and *C. longa* oil in rat endometriosis models. They reported that after the treatment, the volume of ectopic endometriotic foci significantly decreased, and the expression of VEGF was lower in comparison with controls.

*C. longa* extracts arrested endometriosis in dose-dependent manner. It inhibited the expression of MMP-9 and TNF-α, and increased the levels of TIMP-1 [160]. Furthermore, *C. longa* increased the activity of IL-10-1082 A, an anti-inflammatory cytokine gene promoter [161]. Pro-apoptotic activity of *C. longa* has been demonstrated through the enhancement of caspase-3, 9, cytochrome c and Bax [162].

A recent study [163] reported that daily administration of 200 mg/kg of the ethanolic extract of *C. longa* to endometriosis rat models decreased the oxidative stress. After the administration of the ethanolic extracts, the activity of SOD, CAT, and GPx significantly increased, while plasma MDA was lowered. Furthermore, according to Kuo et al. [164], *C. longa* regulated the expression of COX-2, iNOS, CAT, and NO, exerting both anti-oxidative and anti-inflammatory effects.

#### 4.2.7. *Prunella vulgaris*

*P. vulgaris* belongs to Lamiaceae family. It has been used in European Medicine during the 17th century, due to its capacity to reduce fever. In China, it was also employed in folk medicine as antipyretic. This medicinal plant contains flavonoids, rosmaric acid, oleanolic acid, triterpenoids, prunelline, and tannins [165]. It displayed anti-estrogenic effects both in vitro and in vivo, and reduced the surface of endometriotic xenografts, being efficient in the management of estrogen-dependent disorders, such as endometriosis or uterine cancer [166].

*P. vulgaris* was also reported to possess anti-apoptotic effects. It increased the expression of caspase-3 and Bax, and decreased the expression of Bcl-2 [167]. As anti-inflammatory agent, *P. vulgaris* acted through the inhibition of p38 MAPK/ ERK signaling pathway and regulated TNF-α-induced expression of adhesion molecules [168]. Considering all these findings, *P. vulgaris* could be a promising therapeutic alternative for the management of endometriosis.

#### 4.2.8. *Sparganium stoloniferum*

*Rhizoma sparganii* represents the rhizome of *S. stoloniferum*, which is a Traditional Chinese Medicine, used for thousands of years for the treatment of bad stomach [169]. However, preliminary studies of *R. sparganii* suggested its pharmacological mechanisms also involved in angiogenesis and endocrine functions.

Sun et al. [170] investigated the effects of *R. sparganii* in pregnant rodents. They reported that *R. sparganii* significantly lowered the levels of FGF-1 and VEGF, and the expression of ER-alpha was inversely proportional to FGF-1. In these conditions, *R. sparganii* demonstrated its anti-angiogenic and anti-estrogenic activities. SpaTA is a novel polysaccharide isolated from the water extraction of *R. sparganii.* It induced apoptosis in breast cancer cells via caspase-3, 8, and 9 signaling and modulated estrogen signaling. Furthermore, SpaTA regulated the expression of ER-alpha and its nuclear translocation [171]. Topological analysis demonstrated that major targets of *S. Stoloniferum* include IL-8, EGFR, TNF, and VEGFA, which are several of the causal genes of endometriosis [22].

Taken together, all these results indicated that *S. Stoloniferum* could be an efficient therapeutic agent for the management of estrogen-dependent conditions, including endometriosis.

#### 4.2.9. *Salvia miltiorrhiza* (Danshen)

*S. miltiorrhiza* is a perennial Chinese medicinal plant, included in the Lamiaceae family. Since ancient times, dried roots of *S. miltiorrhiza,* also known as Danshen, have been used for the management of cerebrovascular and cardiovascular pathologies [172]. The main bioactive compounds of Danshen are represented by tashinone I and II, cryptotanshinone and salvianolic acid. They possesses a wide range of beneficial pharmaceutical effects, such as anti-inflammatory, antioxidant, antitumoral, and antimicrobial properties [173].

In rat models, after the administration of tanshinone IIA, the growth of the ectopic endometrial tissue was significantly decreased, and the mechanical hyperalgesia was reduced. Moreover, the miRNA levels of angiotensinogen and angiotensin II were lowered in dorsal root ganglion neurons [174]. Tashinone IIA also increased the expression of Bax, a pro-apoptotic protein, and inhibited Bcl-xl and Bcl-2, two anti-apoptotic proteins [175]. Furthermore, this bioactive molecule demonstrated its antiangiogenic properties, by modulating VEGFA/HIF1-α signaling pathway and altering the MMP-2/TIMP-2 ratio in endothelial cells [176]. Tashinone I inhibited the formation of PGE2 from LPS-induced RAW macrophages, without affecting COX-2 expression and activity [177].

Zhou et al. [178] investigated the effects of *S. miltiorrhiza* on Sprague-Dawley rats. After the administration of *S. miltiorrhiza* extracts, serum CA-125 and the levels of IL-18 and TNF-α in the peritoneal fluid significantly were decreased, while peritoneal fluid levels of IL-13 increased, proving the anti-inflammatory properties of this medicinal herb. *S. miltiorrhiza* also showed strong antioxidant potential. Its constitutive polysaccharides increased the activity of GPx, SOD, CAT, MDA, and decreased MDA production [179]. In addition, a recent study pointed out that hydrophilic extract of *S. miltiorrhiza* remarkably enhanced SOD, GSH, and PONase levels [180].

#### 4.2.10. *Paeonia lactiflora* (Chishao)

*Paeonia lactiflora* or Bái Sháo is a Chinese herbal medicine used for more than 1200 years, as a decoction of the dried roots. It is known for being useful in the treatment of dysmenorrhea, muscle cramps, fever, and rheumatoid arthritis. Paeonin is one of the most abundant bioactive compounds from *P. lactiflora,* and its pharmaceutical effects were investigated in vitro and in vivo. *P. lactiflora* extracts inhibited the production of PGE2, leukotriene B4 and NO in animal models. Furthermore, it induced the proliferation of lymphocytes, the differentiation of T helper and T suppressor lymphocytes, and inhibited the delayed-type hypersensitivity in immuno-activated animal models [181]. Paeoniflorin is another major compound of *P. lactiflora.* In postoperative mice, it significantly decreased the pain, via TLR4/MMP-9/MMP2/IL-1β signaling pathway [182].

Endometriosis is a progressive disease, and in some cases, it has a cancer-like behavior. *P. lactiflora* also possesses antitumor properties. Zhang et al. showed that *P. lactiflora* significantly decreased the proliferation of cancerous endometrial cells, in a dose-dependent manner, via the activation of NF-κB and MAPK signaling pathways [183]. Moreover, paeoniflorigenone, induced antiproliferative effects in tumor cell lines and enhanced apoptosis [184].

These results suggest that *P. lactiflora* may induce anti-inflammatory, immunomodulatory effects, and pro-apoptotic effects, and could be a beneficial alternative for the management of endometriosis.

#### 4.2.11. *Viburnum opulus*

*V. opulus,* or European cranberry, belongs to Adoxaceae family and it is usually included in daily diet in juices, jelly, marmalade or jam [185]. It contains high amounts of anthocyanin and exerts potent antioxidative properties. In addition, these fruits possess anti-inflammatory, antimicrobial, antiviral, and antialgic properties [186].

In rat models with surgically-induced endometriosis, *V. opulus* significantly decreased the volume of endometriotic lesions, and lowered the serum levels of IL-6, VEGF and TNF-α [187]. It also possesses strong antioxidant capacity. According to Zayachkivska et al., *V. opulus* induced increased generation of NO, SOD, CAT, and suppressed MDA level and lipid peroxidation process [188].

#### 4.2.12. *Cyperus rotundus*

*C. rotundus* or “Nagarmotha”, is a medicinal plant used in China, India and Japan due to its beneficial effects against stomach disorders, spasms or bowel irritation. It has a high concentration of bioactive compounds, such as flavonoids, ascorbic acids, and phenolic acids. The rhizome of this herb exerts anti-inflammatory, analgesic, antioxidative, and antipyretic properties. In vitro, *C. rotundus* extracts prevented lipid peroxidation, thus demonstrating that this plant could be a potential source of antioxidants [189]. It is also widely used as anti-inflammatory agent. In endometriosis, *C. rotundus* decreased inflammation by mainly targeting the following pathways: HIF-1 signaling pathway, TNF and MAPK signaling pathway [22]. Moreover, it showed cytotoxic and pro-apoptotic effects on endometrial and ovarian tumoral cells [190].

#### 4.2.13. *Euterpe oleracea*

*E. oleracea* is also known as Acai palm, and belongs to Arecaceae family. Usually, Acai berries are macerated with water, separated from their seeds and consumed as a purple-colored beverage [191]. Due to its antioxidant properties, Acai pulp is consumed as functional food.

Machado et al. [192] investigated the effects of *E. oleracea* on rat models with endometriosis. After the treatment, they observed a significant decrease of the endometriotic lesions surface. Furthermore, *E. oleracea* demonstrated its anti-inflammatory properties, by decreasing the levels of VEGF, PGE2, MMP-9, and COX-2. Through the activation of caspase-3, *E. oleracea* induced apoptosis and decreased cellular proliferation [193].

Velutin is a strong anti-inflammatory flavonoid isolated from the acai berries pulp. It was demonstrated that velutin reduces the production of TNF-α and IL-6, secondary to NF-kB inhibition. Velutin also blocked the degradation of NF-κB inhibitor and JNK phosphorylation [194].

*E. oleracea* possesses strong antioxidant properties. According to Zhou et al. [195] it lowers the oxidative stress by decreasing MDA and Nrf2 protein expression, and increasing the levels of SOD and GSH.

#### 4.2.14. Other Natural Products for Endometriosis Management

Pycnogenol^®^ is a natural antioxidant, a bark extract from *Pinus pinaster.* It contains a wide range of phenolic compounds such as taxifolin, catechin and procyanidins, and in the last years, it attracted the attention of many researchers in the field of endometriosis treatment, due to its strong free radical-scavenging activity against ROS [196]. Furthermore, Pycnogenol^®^ is able to increase the efficiency of oral contraceptives for the treatment of endometriosis-related chronic pelvic pain [197]. According to Kohama et al. [198], Pycnogenol^®^ reduced the pain scores without influencing the menstrual cycles pattern and the estrogen levels of endometriosis patients, and decreased serum levels of CA-125. This natural product, induced caspase-independent apoptosis and increased the antioxidant capacity in plasma [199].

Jing Tong Yu Shu is a traditional Chinese medicine with multiple beneficial effects. It consists of eleven herbs: *Angelica sinensis, Asarum sieboldii, Cortex cinnamom, Cornus officinalis, Corydalis yanhusuo, Dioscorea polystachya, Foeniculi Fructus, Faeces trogopterori, Paeonia lactiflora Pallas, Radix cyathulae*, and *Whitmania pigra Whitman*. Zhang et al. [200] reported that after four weeks of treatment with Jing Tong Yu Shu in endometriosis rat models, the volume of the ectopic endometrial tissue foci recorded a significant decrease. In addition, this traditional medicine significantly inhibited the secretion of IL-1β and IL-6 in the peritoneal fluid.

*Urtica dioica* roots and leaves have been used by ancient times for the treatment of menstrual hemorrhage, eczemas or rheumatism-related complaints. It has been demonstrated that hexane, ethyl acetate, and methanol extracts of *U. dioica* exhibited promising effects in endometriosis rat models. After the administration of this extracts obtained by the aerial parts of *U. dioica*, the levels of IL-6, TNF-α, and VEGF were significantly lowered [201].

*Zingiber officinale,* also known as Ginger, is a widely used root in Asian cuisine; also, it is used as medicine, due to its multiple beneficial pharmaceutical properties. Shagaols, are bioactive compounds found in dried ginger, which possess anti-inflammatory, anticancer, antioxidant, and neuroprotective activities [202]. Wang et al. [203] explored the effects of 6-shogaol on endometriosis rats models and described the main anti-inflammatory pathways of this bioactive compound. After one month of oral gavage with 6-shagol, the surface of the endometriotic lesions was reduced, the histological analysis suggesting atrophy and major regression. 6-shagol modulated the expression of VEGF and VEGFR-2 and down-regulated NF-κB signaling. Furthermore, pro-inflammatory factors such as IL-6, IL-1β, PGE2 and NO were suppressed. In these conditions, *Z. officinale* extracts seem to be promising therapeutic agents for the management of endometriosis.

Table 1 summarizes the main biological effects of herbal medicine against endometriosis and their mechanisms of action.

## 5. Conclusions

Despite being a benign gynecological condition, endometriosis continues to be a debilitating disease for young women in reproductive age, due to the symptoms that it develops. Chronic pelvic pain, vaginal bleeding, infertility, and malignant transformation are the most widely encountered symptoms. Considering the lack of specific symptoms and specific biomarkers for endometriosis diagnosis, the accurate diagnosis could be delayed for years, and the gold-standard remains surgery followed by histopathological exam. Medicinal herbs and their bioactive compounds exhibit anti-angiogenic, antioxidant, sedative and pain-alleviating properties and the beneficial effects recorded until now encourage their use for the management of endometriosis.

In this paper, we described the pharmaceutical activity of 13 phytochemicals and 17 medicinal plants, including those used by centuries for the Traditional Chinese Medicine. All these natural compounds are able to interact with multiple biological processes, resulting in the alleviation of endometriosis-related complaints. However, future human randomized clinical trials would be helpful in order to achieve more conclusive results about the efficiency of herbal medicine as alternative therapy in endometriosis.

In our opinion, the administration of herbal therapy to the patients with endometriosis could be a topic of great interest for the supporters of natural therapies. Low cost, decreased number of side effects, increased bioavailability, as well as the favorable results reported until now are several of the arguments that could support the naturopathic treatment of this pathology. However, considering that the etiopathology of endometriosis is so debated and not fully elucidated, it requires caution in deciding the therapeutic management. Moreover, the lack of human clinical trials represents a major disadvantage for this therapy, although in vitro and in vivo studies are very promising.

Therefore, we consider medicinal plants and phytochemicals excellent adjuncts in the treatment of endometriosis, but they are currently insufficient as unique therapeutic tools. The decision to consume such dietary supplements must be made with a doctor.

## Figures and Tables

**Figure 1 plants-10-00587-f001:**
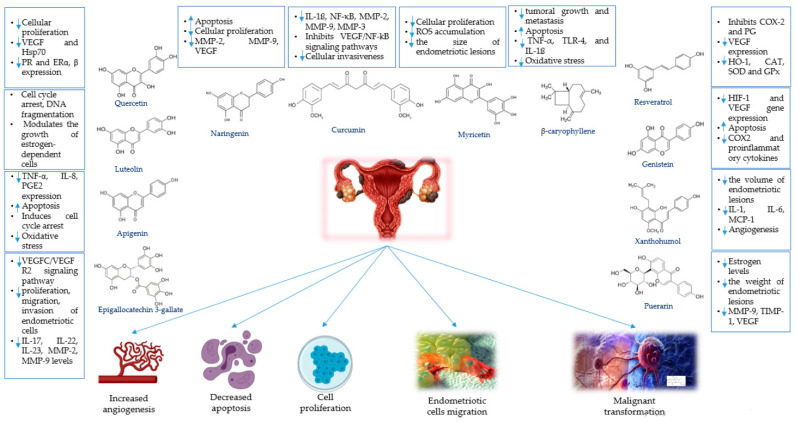
Endometriosis pathogenic mechanisms and the main mechanisms of action of various phytochemicals.

**Table 1 plants-10-00587-t001:** Medicinal herbs and their effects against endometriosis.

Medicinal Plant		Experimental Model	Biological Effects	Molecular Mechanisms	Reference
*Angelica sinensis*	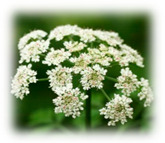	Rats with surgically induced endometriosis (human endometriotic cells)	Reduced the number of endometriotic lesions Anti-inflammatory effects	Decreased the levels of IL-18, TNF-α, MMP-2, MMP-9, and CA-125 Increased the levels of IL-13	[140]
*Achillea biebersteinii*	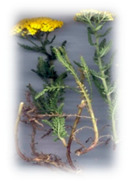	Rats with surgically induced endometriosis Mouse pain models	Reduced the volume of endometriotic lesions Inhibited the formation of peritoneal adhesions Reduced endometriosis-related dysmenorrhea (antinociceptive effects) Anti-oxidative effects	Decreased the levels of IL-6, TNF-α, VEGF Inhibited the nociceptive receptors and its effects were blocked by Atropine	[143,144,145]
*Artemisia princeps*	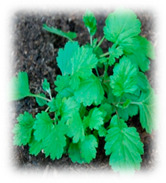	Human endometriotic cells	Increased apoptosis in endometriotic cells Anti-proliferative effects Anti-inflammatory effects Anti-oxidative effects	Inhibited the expression of Bcl-2, Bcl-xL, XIAP, caspase 3, caspase 8 and caspase 9 Induced cell cycle arrest in G2/M phase Decreased the levels of of TNF-α, ICAM-1, IL-1ß, and VCAM-1 Suppressed RO, NO and iNOS expression in LPS-induced macrophages	[148,149,150,151]
*Allium sativum*	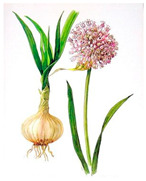	Human endometriotic cells	Reduced cellular proliferation Induced apoptosis Anti-angiogenic activity	Reduced the expression of VCAM-1 and ICAM-1 Increased the activity of caspase-3 and Bax expression Decreased VEGF and VEGFR expression	[151,153]
*Astragalus membranaceus*	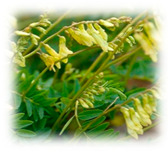	Rats with surgically induced endometriosis	Decreased the volume of endometriotic lesions Anti-estrogenic and anti-progestative effects Anti-inflammatory effects	Decreased the concentrations of IL-2, TNF-α, estrogen and progesterone	[156]
*Curcuma longa*	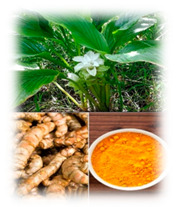	Rat with surgically induced endometriosis	Decreased the volume of endometriotic lesions Anti-angiogenic effects Anti-inflammatory effects Pro-apoptotic effects Antioxidative effects	Suppressed the expression of VEGF, MMP-9, TIMP-1 and TNF-α Increased the activity of IL-10-1082 A Increased the activity of caspase-3, caspase-9 and Bax Increased serum levels of SOD, CAT and GPx	[159,160,161,162,163]
*Prunella vulgaris*	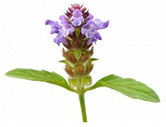	Xenograft mice	Anti-estrogenic effects	Inhibited the expression of CYR61, CYP1A1, CYP1B1 Induced decreased proliferation of ER	[166]
*Rhizoma sparganii*	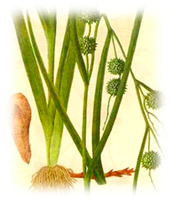	Pregnant rodents	Anti-estrogenic effects Anti-angiogenic effects	Decreased the levels of FGF-1, VEGF, and ER-α	[170]
*Salvia miltiorrhiza*	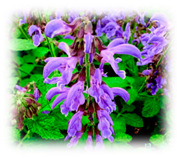	Rats with surgically induced endometriosis	Inhibited the growth of endometriotic lesions Reduced hyperalgesia Anti-inflammatory effects	Decreased miRNA levels of angiotensinogen and angiotensin II in dorsal root ganglion neurons Decreased the levels of IL-18, TNF-α in the peritoneal fluid	[174,178]
*Paeonia lactiflora*	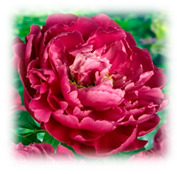	Tumoral endometrial cells	Anti-proliferative effects	Activated NF-κB and MAPK signaling pathways	[183]
*Viburnum opulus*	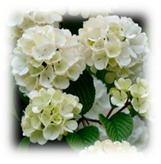	Rats with surgically induced endometriosis	Decreased the volume of the lesions Exhibited antioxidant capacity	Decreased IL-6, VEGF and TNF-α Increased the levels and activity of NO, SOD, CAT	[187,188]
*Cyperus rotundus*	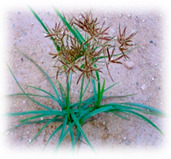	Human endometrial cells	Anti-inflammatory effects	Targeted HIF-1 signaling pathway, TNF and MAPK signaling pathway	[22]
*Euterpe oleracea*	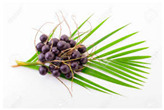	Rats with surgically induced endometriosis	Decreased the surface of the lesions Anti-inflammatory and pro-apoptotic effects	Decreased the level of VEGF, PGE2, MMp-9, COX-2 Activated caspase-3	[193]
*Pinus pinaster*	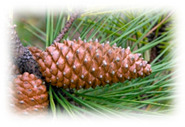	Human clinical trial	Reduced the chronic pain scores Pro-apoptotic effects	Decreased the levels of CA-125, modulated caspases activity	[198,199]
*Urtica dioica*	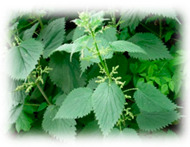	Rats with surgically induced endometriosis	Anti-inflammatory effects Anti-angiogenic effects	Decreased the level of, the levels of IL-6, TNF-α Suppressed VEGF	[201]
*Zinguber officinale*	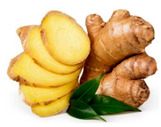	Rats with surgically induced endometriosis	Induced the atrophy and regression of endometriotic lesions Anti-angiogenic effects Anti-inflammatory effects	Decreased the expression of VEGF and VEGFR Down-regulated NF-κB signaling Decreased the levels of IL-6, IL-1β, PGE2, NO	[203]

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
