# Peer review of "An Overview on the Conservative Management of Endometriosis from a Naturopathic Perspective: Phytochemicals and Medicinal Plants"

_plants, 2021, doi:10.3390/plants10030587_

Round 1

Reviewer 1 Report

The paper entitled: "An Overview on the Conservative Management of Endometriosis from a Naturopathic Perspective: Phytochemicals and Medicinal Plants" is very interesting from a scientific-medical point of view and brings a welcome synthesis of many natural alternatives / adjuvants in endometriosis. Unfortunately for patients, for now these alternatives are not a recommended solution, due to the lack of more complex studies, but they represent a hope for the future. I recommend publishing this review with some additions / improvements by the authors.

1. Did you state in Chapter 4.1 that an increasing number of women are looking for alternative options, such as herbal medicine, which is a promising strategy for managing endometriosis, but should this treatment be recommended by a specialist or pharmacist?

  1. Have in vivo EGCG studies been performed in human patients?
  2. How were they chosen and how was the order of presentation of the active principles established? It should be systematized, perhaps according to the basic structure: flavones, flavanones, isoflavonoids: - Apigenin, myricetin are flavones like luteolin and chrysin - Genistein and puerarin are two isoflavonoids - other polyphenols: resveratrol, Epigallocatechin-3-gallate, curcumin, etc.
  3. Plants named in Latin are written in italics (example: Camelia sinensis).
  4. Rhizoma sparganii is the plant product = the rhizome of Sparganium stoloniferum. Please replace in title 4.2.8.
  5. Viburnum opulus belongs to the Adoxaceae family.
  6. Zinguber officinale - Please correct!
  7. Is Jing Tong Yu Shu an herbal medicine? Is his composition known?
  8. A small image with the medicinal plant could be inserted in the table.

Author Response

Dear Sir or Madam,

According to your suggestions, we made the following modifications:

Q1. Did you state in Chapter 4.1 that an increasing number of women are looking for alternative options, such as herbal medicine, which is a promising strategy for managing endometriosis, but should this treatment be recommended by a specialist or pharmacist?

A: No, only a clinical physician should recommend the naturopathic treatment of endometriosis. We discussed this aspect in the Conclusion and in chapter 4.1.

Q2. Have in vivo EGCG studies been performed in human patients?

A: No, there are no human clinical trials conducted until now in this field.

Q3. How were they chosen and how was the order of presentation of the active principles established? It should be systematized, perhaps according to the basic structure: flavones, flavanones, isoflavonoids: - Apigenin, myricetin are flavones like luteolin and chrysin - Genistein and puerarin are two isoflavonoids - other polyphenols: resveratrol, Epigallocatechin-3-gallate, curcumin, etc.

A: We ordered all the phytochemicals alphabetically.

Q4. Plants named in Latin are written in italics (example: Camelia sinensis).

A: We wrote all the plants named in Latin with italics

Q5. Rhizoma sparganii is the plant product = the rhizome of Sparganium stoloniferum. Please replace in title 4.2.8.

A: We made the replacement

Q6. Viburnum opulus belongs to the Adoxaceae family.

A: We made this modification

Q7. Zinguber officinale - Please correct!

A: We made this modification

Q8. Is Jing Tong Yu Shu an herbal medicine? Is his composition known?

A: Yes, JTYS is a mixture of 11 herbal medicines. We included the composition of JTYS in our paper. (Line 816)

Q9. A small image with the medicinal plant could be inserted in the table.

A: We inserted images with the medicinal plants in the second column of the table

Thank you very much for your valuable comments!

Respectfully,

Andreea Balan

Reviewer 2 Report

The authors analyzed in this review paper all the studies of the last 20 years of literature. Their work focused on the efficiency of medicinal plant extracts and phytochemicals in the conservative management of endometriosis and endometriosis-related symptoms. In my opinion, this paper is nice, well organized, and the flow of information is very lucid. I consider that it would be of great interest to the researchers and the readers of Plants Journal.

However, I make some remarks to further improve the quality of the manuscript:

  • Do the authors believe the dietary supplementation with medicinal plant extracts and phytochemicals will be a sufficient measure against endometriosis? Please discuss this aspect
  • Please rewrite the conclusions by giving a more personal note to this chapter. The authors should express a personal opinion on the use of plant extracts for the treatment of endometriosis
  • Figure 1 and Table 1 synthesizes in an excellent manner all the information discussed in the Results section. Please confirm that Figure 1 is original, and all the chemical structures included in Figure 1 are drawn by the authors
  • In line 158 ''increases'' should be replaced with ''increased''
  • In line 167 ''inflammation'' should be replaced with ''inflammatory''
  • The following phrase is a bit confusing: '' the phagocytic ability of macrophages is mediated by the secretion of matrix metalloproteinases (MMP), that are enzymes capable to destroy the organization of extracellular matrix, and by the expression of several macrophages surface receptors, which are able to promote the dissolution of cell debris''. Please revise
  • In line 185 ''peritoneal fluid MMP-2 expression in affected women'' should be replaced with ''the expression of MMP-2 in the peritoneal fluid of the affected women''
  • There are also few spelling mistakes and language errors that should be revised

Author Response

Dear Sir or Madam,

According to your suggestions, we made the following modifications:

  • We completed the Conclusion section with or personal opinion regarding herbal therapy in the management of endometriosis
  • All the chemical structures from Figure 1 are drawn by the authors using KingDraw.
  • Spelling mistakes and language errors were revised.

Thank you for your comments!

Respectfully,

Andreea Balan

Reviewer 3 Report

The MS is well written and I suggest a few minor corrections:

Page 4/41 – line 172 – destroy - spelling

Pafe 6/41 – line 290 - Camelia sinensis – Italics

Page 9/41- line 451- decreased – spelling

Page 13/41 – line 605 – please rephrase the sentence

Page 14/41 – line 636 -   add space after A in A.sativum

Page 22/41 – line 881 – Review instead of paper

Interleukin spelling in Abbreviations

Author Response

Dear Sir or Madam,

According to your suggestions, we revised all the spelling errors. Also, we introduced images of the medicinal plants in the second column of Table 1, and we completed the Conclusion section by giving this chapter a more personal note.

Thank you for your valuable comments!

Respectfully,

Andreea Balan
